# Effectiveness of Blood Flow Restriction on Functionality, Quality of Life and Pain in Patients with Neuromusculoskeletal Pathologies: A Systematic Review

**DOI:** 10.3390/ijerph20021401

**Published:** 2023-01-12

**Authors:** Álvaro Jesús Reina-Ruiz, Jesús Martínez-Cal, Guadalupe Molina-Torres, Rita-Pilar Romero-Galisteo, Alejandro Galán-Mercant, Elio Carrasco-Vega, Manuel González-Sánchez

**Affiliations:** 1Department of Physiotherapy, Faculty of Health Sciences, University of Málaga, 29071 Málaga, Spain; 2Department of Nursing, Physiotherapy and Medicine, Faculty of Health Sciences, University of Almería, 04120 Almería, Spain; 3Institute of Biomedicine of Málaga (IBIMA), 29010 Málaga, Spain; 4Institute of Biomedicine of Cádiz (INIBICA), 11009 Cádiz, Spain; 5MOVE-IT Research Group, Department of Nursing and Physiotherapy, Faculty of Health Sciences, University of Cádiz, 11009 Cádiz, Spain

**Keywords:** blood flow restriction, pain, quality of life, exercise, neuromusculoskeletal disorders

## Abstract

Background: Blood flow restriction is characterized as a method used during exercise at low loads of around 20–40% of a repetition maximum, or at a low-moderate intensity of aerobic exercise, in which cuffs that occlude the proximal part of the extremities can partially reduce arterial flow and fully restrict the venous flow of the musculature in order to achieve the same benefits as high-load exercise. Objective: The main objective of this systematic literature review was to analyze the effects of BFR intervention on pain, functionality, and quality of life in subjects with neuromusculoskeletal pathologies. Methods: The search to carry out was performed in PubMed, Cochrane, EMBASE, PEDro, CINHAL, SPORTDiscus, Trip Medical Database, and Scopus: “kaatsu” OR “ischemic training” OR “blood flow restriction” OR “occlusion resistance training” OR “vascular occlusion” OR “vascular restriction”. Results: After identifying 486 papers and eliminating 175 of them due to duplication and 261 after reading the title and abstract, 50 papers were selected. Of all the selected articles, 28 were excluded for not presenting a score equal to or higher than 6 points on the PEDro scale and 8 for not analyzing the target outcome variables. Finally, 14 papers were selected for this systematic review. Conclusions: The data collected indicate that the blood flow restriction tool is a therapeutic alternative due to its effectiveness under different exercise modalities. The benefits found include decreases in pain thresholds and improvement in the functionality and quality of life of the neuro-musculoskeletal patient during the first six weeks. However, the results provided by this tool are still not clear for medium- and long-term interventions.

## 1. Introduction

Currently, there are different strategies to restrict blood flow before or during exercise seeking to improve performance such as remote ischemic preconditioning or blood flow restriction (BFR).The term remote ischemic preconditioning refers to a type of blood flow restriction focused on sport performance where the cuffs are inflated and deflated during sets [1]. Second, BFR could be considered as a training adjuvant that in recent years has been used during physical exercise at low intensities during strength training (around 20–40%) or at low or medium intensities during aerobic exercise with the intention of improving cardiopulmonary capacities. In [2], in which cuffs that occlude the proximal part of the extremities can partially reduce arterial flow and fully restrict venous flow of the musculature in order to achieve the same benefits as high-load exercise [2,3]. Several theories attempt to justify the results obtained from BFR intervention. One of these theories considers that the metabolic effects of induced hypoxia cause a higher amount of lactate [4], reactive oxygen free radicals, and nitrogen oxide [5], which lead to an increase in protein synthesis and the recruitment of muscle fibers as well as to a decrease in proteolysis [6]. On the other hand, the effects of BFR intervention may be due to neuromuscular and hormonal reactions, given that the cuff pressure, between 50 and 230 mm Hg, induces a greater muscle activation [7,8], and even a greater secretion of the growth hormone IGF-1 depending on the pressure applied by the cuff [9].

In rehabilitation, the working methodologies employed are heterogeneous in terms of the occlusion measurement used, arterial occlusion pressure or limb occlusion pressure, differences in the material used, cuff amplitude, amount of pressure applied, and the type of training [10,11].

Currently, it has been analyzed whether BFR intervention is capable of directly influencing other variables such as pain, functionality, and quality of life, obtaining favorable results that indicate that this technique can improve, above all, the intensity of pain and the ability to perform activities of daily living [12]. Although there are many systematic reviews that analyze the effect of BFR intervention [3,13,14,15,16,17,18,19,20,21], no systematic review has been found that analyzes the effect of BFR intervention on subjective and objective variables in the short-, medium-, and long-term in patients suffering from neuromusculoskeletal pathologies. Thus, to date, and to the best of our knowledge, no review has analyzed how BFR intervention can directly contribute to these variables in the rehabilitation area. The main objective of this systematic literature review was to analyze the effects of BFR intervention on pain, functionality, and quality of life in subjects with neuro-musculoskeletal pathologies.

## 2. Methods

A systematic review was carried out using the guidelines of the PRISMA checklist system.

### 2.1. Search Strategy

The search to carry out this systematic review was performed in different Health Sciences databases (PubMed, Cochrane, EMBASE, CINHAL, SPORTDiscus, Trip Medical Database, Scopus) using the terms: “kaatsu” OR “ischemic training” OR “blood flow restriction” OR “occlusion resistance training” OR “vascular occlusion” OR “vascular restriction”. The search included all the articles published up to 31 October 2022, which were subsequently evaluated [22].

### 2.2. Selection of Documents

This study included articles that were designed as a randomized clinical trial, whose objective was to evaluate therapeutic exercise in combination with blood flow restriction on subjects with neurological and musculoskeletal pathologies, and that were published up to 31 October 2022. For inclusion, the studies had to present at least one intervention or control group in which the BFR intervention was applied.

Studies not published in master theses, case reports, reviews, cross-sectional studies, and cohort studies were excluded. The PEDro scale considers that studies with a score of less than 6 points present a sufficient or poor methodological quality, thus they were excluded [23].

### 2.3. Selection Method

In parallel and blinded, two researchers conducted a search of the scientific literature in the different databases and selected the documents applying the inclusion and exclusion criteria set out above. After reviewing the articles in the different databases, all those that were duplicates were discarded. In the case of discrepancy, the paper selection was performed by a third blinded researcher. Once the articles were selected, their internal validity was assessed using the PEDro scale.

### 2.4. Methodological Quality

The methodological quality of the randomized clinical trials was assessed using the PEDro rating scale. This scale consists of 11 questions, of which 10 were scored by a dichotomous answer (Yes/No), which was awarded on the basis of meeting the requirements of the specified section. The unscored question was discarded due to influences of external validity and not internal or statistical validity of the trial [24].

### 2.5. Outcomes

The outcome variables involved in the analysis of the selected studies were pain, functionality, and quality of life. The pain variable should be understood from a more global perspective such as the biopsychosocial one apart from purely biological damage such as nociceptive or neuropathic pain [25].

Functionality is understood as the individual’s ability to move around in different types of environments with total safety and independence in order to be able to perform all of the tasks that are part of the person’s activities of daily living in any context [26]. On the other hand, quality of life is defined as the state of life perceived by the individual regarding the impact of their pathology and the treatment performed on their disability and daily functionality [27].

Likewise, a time continuity model was established for the analysis of the results of the study. In order to compare the results between the different studies, the results were standardized on a scale from 0 to 100.

Data were collected at the baseline, short- (0–6 weeks), medium- (6–12 w), and long-term (12–24 w), and the follow-up period as the post-treatment period.

## 3. Results

After identifying 486 papers in the different databases, and eliminating 175 of them due to duplication and 261 after reading the title and abstract, 50 papers were selected. Of all the selected articles, 28 were excluded for not presenting a score equal to or higher than 6 points on the PEDro scale and 8 for not analyzing the target outcome variables. Finally, 14 papers were selected for this systematic review (Figure 1).

Table 1 shows the structural characteristics of the different studies selected. The total sample of all the studies included in this study was 533 patients (Table 1) with a minimum of 22 patients [28] and a maximum of 79 [29], with a mean age of 44.5 years, ranging from 14 [30] to 75 years [28]. The different types of interventions employed were aerobic exercise [31] and strength exercise [29]. In the selected papers, the session frequencies ranged from one to three sessions per week [31,32], while the intervention duration ranged from one session [32] to 16 weeks [29].

### 3.1. Functionality (Objective Outcomes)

Table 2 shows all the studies that assessed the functionality outcome from an objective point of view. For the outcome variable “time”, a mean of 6.87 s with a minimum of 5.5 s and a maximum of 8.25 s was recorded in the Time Up and Go Test, showing similarities between the results of both studies that used this tool [33,40]; in the variable “distance”, a mean of 367 m was obtained, with a minimum of 120 m and a maximum of 614 m, showing differences between the beginning and the end of the intervention of the two studies that employed this instrument [31,38]; in the variable “repetitions”, a mean of 14.65 repetitions was obtained, with a difference between 10.5 and 18.8 repetitions, showing similar results [33,40]; and in the variables “speed” and “power”, a direct comparison of the results could not be made, since the corresponding studies used different evaluation tools or presented a single study for the outcome variable. In general, all of the studies collected results during the first 3 months of the intervention and did not perform a post-intervention follow-up, except for the studies by Lamberti et al. and Segal et al. [38,41], who continued to collect data at a 3-month follow-up interval.

### 3.2. Functionality (Subjective Outcomes)

Table 3 presents the results related to functionality extracted from the selected studies. Up to 10 assessment tools were used such as the Short Physical Performance Battery, Late Life Function and Disability Instrument [35], International Knee Documentation Committee [30,37], Lysholm Knee-Scoring Scale, Tegner Activity Scale, Knee Osteoarthritis Outcome Score, Lower Extremity Function Scale [37], Inclusion Body Myositis Functional Rating Scale [28], Health Assessment Questionnaire [28,40], and Berg Balance Scale [38]. In the different tools, the International Knee Documentation Committee stood out, with a mean score of 26.88 and values between 6.08 and 42.69 points, in which the results of both studies reflect the disparity [30,37], which occurs in the Health Assessment Questionnaire, which presents a mean value of 1.03 points, with a minimum of 0.16 and a maximum of 1.9 points [28,40]. The results of all of the studies were collected in the first 3 months of the intervention, since most of them did not present a follow-up like the studies by Lamberti et al. and Segal et al. [38,41].

### 3.3. Pain

Regarding the outcome variable “pain”, Table 4 shows eight assessment tools. Among the different measurement instruments, the questionnaires with the highest frequency of use are the Kujala Patellofemoral Score, with an average value of 44.33 between 0 and 88.7 points [41,42,43], and the Visual Analogue Scale with an average score of 42.85, a minimum of 0 points, and a maximum of 85.7 points [29,35,40]. All of the studies collected data during the first 3 months of the intervention, and only five studies carried out a follow-up 3 months after the intervention.

### 3.4. Quality of Life

Data were collected from four assessment tools (Table 5), showing less heterogeneity among the results. Likewise, the results obtained in the different studies were collected up to 3 months after the intervention (follow-up), and only two studies [28,38] evaluated the data after the intervention in up to 3 months of follow-up, where the Short Form-36 Health Survey stands out with a mean of 57.35 points, whose minimum and maximum values were 14.7 and 100 points, respectively, although with differences in the collection of results, recording them globally or by areas, and showing a lack of homogeneity among the results [33].

## 4. Discussion

Based on the results of this systematic review analyzing the effects of BFR intervention and different exercise methodologies, it appears that there is a favorable trend of BFR intervention in the short-term for each of the aforementioned variables despite the great heterogeneity of the population groups. This tends to normalize in the medium- and long-term, although there are not enough studies to be consistent. In addition, after analyzing the results presented in this systematic review, it was observed how the use of BFR intervention, as an adjuvant methodology during treatment in patients with neurological or musculoskeletal pathologies, achieved greater or similar benefits compared to high-load exercises in the initial processes of recovery due to the metabolic stress and the lower mechanical stress it induced. In this way, high intensity would be simulated through low load, as long as the treated structure allows for the introduction of BFR intervention as a complement to exercise, although there are aspects that must be analyzed specifically depending on the studied variable.

### 4.1. Functionality

#### 4.1.1. Time

In terms of the functional outcome time (Table 6), it was observed that the BFR intervention group improved between 2.9 and 6 s in the sit-to-stand 5 times (STS5), performed by patients with pathologies such as multiple sclerosis or knee arthroscopy [38,43]. This difference in the results may indicate that a lower pressure and walking with BFR intervention obtains better results than higher pressures and strength exercises. However, other factors such as the rating of perceived exertion or the total elimination of pressure between exercises could influence the losses in the metabolic effect induced by the BFR intervention [44,45]. Regarding the other tests, there was a trend in favor of the BFR intervention, with a 2 s difference with respect to the control group in the Four Square Step Test (FSST), and 4.4 s in the Timed Stair Ascent (TSA) [43]. One possible consequence is the metabolic stress that the study subjects were subjected to with this tool rather than the mechanical stress provided by high loads [2]. However, mechanical stress may also play a role, as the control group did not add any strength exercise compared to the BFR intervention group, which may benefit from specific metabolic responses when performing the low-load exercises [46].

In the medium-term, intra-group changes of up to 0.75 s are shown in the Time Up and Go Test [33,40]. These improvements were very similar in all intervention groups, even when using different exercise methodologies such as high loads, low loads, or BFR intervention, except for the control group, which did not perform any intervention. This could indicate that the subjects improved, regardless of the metabolic or mechanical stress produced by the exercises [47]. In the long-term, only one study carried out a short-term follow-up in which the improvements obtained were slightly reduced [38].

#### 4.1.2. Speed

With respect to the speed category, in the short-term, benefits were observed in the Timed 25-Foot Walk Test (T25FW) of 0.12 m per second in favor of the BFR intervention group [38]. This difference may be relevant as a change of 15% could be significant [48] and the fact that the control group walked at a high-intensity level controlled by rating perceived exertion. Nonetheless, the intervention group introduced this intensity by metabolic stress with the BFR intervention, and controlled the rests with a metronome, which may have produced better results at the brain level, especially anticipatory motor control [49]. In self-selected walking velocity (SSWV), both groups improved between 0.46 and 0.49 m per second, which may indicate that subjects with knee arthroscopy obtain the same benefits from BFR intervention and mechanical stimulus of the loads [43]. In the same way, similar results were seen in the 400-m-walk gait speed, where the BFR intervention group presented −0.01 m per second, which might explain why they did not achieve enough adaptations [35], regardless of whether exercises with different loads are used, even with the application of BFR intervention [50].

The lack of medium-term changes shown for the 400-m-walk gait speed tool suggests the same aforementioned before [35,51]. Long-term data were only recorded for short-term follow-up, with a decrease of 0.03 m per second for both groups at T25FW [38]. This suggests that BFR intervention, in combination with functional exercise, is maintained over time and is of interest in populations with neurological pathologies, which may have accentuated corticomotor activation after metronome use [52].

#### 4.1.3. Distance

For this category, short-term changes were found in favor of the BFR intervention, with a 49-m difference in the 6-min-walk test [38]. Nevertheless, the BFR intervention and control groups improved regardless of the type of intervention, although the BFR intervention group showed greater benefits, specifically about 17 more meters, suggesting that it is an interesting tool during aerobic exercise, enhancing its main physiological effect [2]. In a similar way, the modified Star Excursion Balance Test (SEBT) showed better results in each of the categories for the BFR intervention group, although these data cannot be assumed to be due to the application of this tool [37]. However, the origin of this improvement may lie in the improvement of the quadriceps muscle activation deficits shown by patients with anterior cruciate ligament reconstruction, thanks to the corticomotor activation of the BFR intervention; even the effects induced by cross-training can improve the functional deficits in both lower limbs [53,54].

The results shown in the modified SEBT are favorable in the medium-term for the BFR intervention group in both healthy and injured limbs, with mean changes of 10.3% and 10.6% in the anterior axis, 10.8% and 13% in the posteromedial axis, and 8.9% and 11.5% in the posterolateral axis, respectively [37]. The improvements presented in the BFR intervention group suggest that the effects of using this tool with low loads are superior, as they introduce less mechanical intensity. This benefit may be given that, in the early phase, the tissue used for anterior cruciate ligament reconstruction does not have sufficient tolerance and adaptation to high loads [55].

In relation to the long-term, only one study using the 6-min-walk test showed that the changes produced during the short-term follow-up were maintained [38]. Thus, BFR intervention could be used as a complement of training to enhance the effects of therapeutic physical exercise in patients with multiple sclerosis, despite being a relative contraindication in neuropathy or spinal cord injury [8].

#### 4.1.4. Other Objective Functional Outcomes

The timed stand test showed a difference of up to 2.25 repetitions for the experimental group, which performed exercises with high loads [33,40]. This may explain that the mechanical stress and the progression with loads were better in high loads rather than the metabolic stress induced by BFR intervention. Even so, BFR intervention is therefore considered another alternative to improve in patients with rheumatoid arthritis in the short-term [56,57]. Regarding long-term and follow-up, no results were shown.

According to the outcome “power”, this only presented data in the short-term follow-up, showing a mean difference of 53.4 watts in the stair climb power in favor of the control group [41]. One possible cause could be the use of different pressures throughout the trial, and the other might be the degree of occlusion controlled by a Doppler ultrasound, which could not reach the necessary parameters to create adaptations in terms of the metabolic stress required [45].

### 4.2. Subjective Functionality

According to the subjectivity of functionality, the Berg Balance Scale (BBS) showed an increase of two points in both groups in the short-term [38]. This indicates that BFR intervention provides the same benefits as a functional activity, as both work at aerobic thresholds.

In the medium-term, differences of 0.8 points were found in the BFR intervention group for the Short Physical Performance Battery [35]. This datum may suggest that the use of this tool may be beneficial in the early stages, instead of using high loads due to the control and progression in the degree of occlusion between sessions using systolic blood pressure and thigh thickness as a reference [58]. The Health Assessment Questionnaire (HAQ) showed a mean difference of 0.05 points in favor of the BFR intervention compared to doing nothing or training with high loads [40]. It is possible that muscle hypertrophy due to the effect of active hyperemia during BFR intervention sessions is interacting with the benefits obtained or unknown physiological mechanisms [59]. For the Late Life Function and Disability Instrument (LLFDI), data of −0.5 points in a total frequency for the BFR intervention group and one point in total limitation for the control group were achieved, thus both high loads and BFR intervention may provide the same benefits in rheumatoid arthritis [40]. In fact, this may be a consequence of the work proposed in the endurance strength groups, where the aerobic threshold can be reached, which is beneficial for the pathology [56].

Relative to the changes shown between the short- and medium-term, intra-group differences of 13.19 points were found for the BFR intervention group in the International Knee Documentation Committee (IKDC), thus we could assert that the younger the person, the greater the adaptation to this tool [37]. It should be noted that BFR intervention was used with high loads instead of low loads, and with high pressures, generating greater stress on the tissue and at a central level, which can influence muscle fatigue and, in turn, increase muscle thickness [60]. The BFR intervention group improved function by 14.81 points on the Lower Extremity Function Scale (LEFS) and 14.83 on the Lysholm Knee-Scoring Scale as a result of better adaptation to low loads. However, the Knee Osteoarthritis Outcome Score (KOOS) showed benefits in the control group for pain with 10.33 points, and symptoms with 12.33 points, and in the BFR intervention group for activities of daily living with 10.5 points, and quality of life with 14.38 points [37,39]. Despite these data, the differences between high loads or low loads with BFR intervention were not large, providing the same benefits for functionality in anterior cruciate ligament reconstruction.

No long-term data were recorded, as with the Tegner Activity Scale, which does not collect data at any time except for the initial assessment and with homogeneous results [37]. In the short-term follow-up, the BBS did not change independently of the intervention [38]. According to the IBMFRS, the control group decreased both sections by up to 17 points compared to the BFR intervention group, which could be due to the fact that they did not perform any particular activity [28] or that the level of self-efficacy was maintained in those patients who practiced exercise [61]. The same is true for the medium-term follow-up where the control group improved compared to the BFR intervention group by 11 points [39]. This may be due to the loss of BFR intervention properties such as metabolic stress when this adjuvant treatment is withdrawn, despite continued exercise [62].

### 4.3. Pain

For the pain outcome, short-term changes of 5.6 points were shown for the control group in the KOOS [42]. Perhaps the lack of specificity in applying the necessary pressure to occlude the artery is not enough to create adaptations in the subject as well as to work on progression [58]. Furthermore, another study showed changes for the BFR intervention group of 13.2 points in the pain section, up to 30 points in the symptoms and activities of daily living section, 28 points in the quality of life section, and 37.5 points in the sport section [43]. This suggests that strength training, especially low-load strength training in conjunction with BFR intervention, may be vital for patients with knee arthroscopy to increase the pain threshold [63]. Likewise, a mean decrease in pain was found on the Numeric Pain Rating Scale (NPRS) in favor of the BFR intervention group of 2.6 points in single leg squat shallow, 2.7 points in single leg squat deep, and 2 points in the side-down test [32]. The decrease in both groups may have been due to the use of the metronome [64] as it modulates tendon pain and corticospinal control of the muscle, especially in the BFR intervention group, where the perception of pain due to cuff pressure was altered [65]. The same was true for the Kujala Patellofemoral Score, where both groups improved equally by around 20–22 points, with the BFR intervention group being slightly higher by almost two points [34]. One consequence of both groups obtaining similar values may be evidence that an exercise program focused on the hip and knee favors the perception of pain intensity [66].

In the medium-term, there was an average improvement of 22.3–26.6 mm in the Visual Analogue Scale (VAS) [29,40] and almost 13 points in the Kujala Patellofemoral Scale [29] in BFR intervention group. The BFR intervention application was superior to high loads in reducing the sensation of pain in subjects of different ages with knee pathologies such as rheumatoid arthritis, knee osteoarthritis, and patellofemoral pain. This finding might be related to the application of high pressures with the cuff, which induces the hypoalgesia mechanism during exercise, thereby increasing the pain threshold [57]. For the Western Ontario and McMaster Universities Osteoarthritis Index (WOMAC) pain subscale, changes of almost seven points were observed for the control group [35]. Nevertheless, the BFR intervention group did not differ much from the results obtained in the control group, thus performing resistance training at moderate intensity and low-load exercises with BFR intervention obtained the same results for knee osteoarthritis, indicating that BFR intervention is a good alternative to improving strength and functionality with lower pain sensation [67].

Between the short- and medium-term, the Pain Börg Scale reports changes in muscle soreness in the BFR intervention group of 0.25 points for the injured leg and 0.7 points for the uninjured leg, while the experimental group improved in the knee soreness scores by 1.3 points for the injured leg and 3.3 points for the uninjured leg [36]. This indicates that the BFR intervention and high-load groups are compatible in accelerating the recovery process in patients with an anterior cruciate ligament reconstruction and decreasing the level of pain. The BFR intervention group would generate this through cuff pressure, metabolic stress, and lower loads, while the control group would use higher mechanical stress [68].

In the long-term, no data were collected for the pain variable, although data were collected for the short- and medium-term post-intervention follow-up, all groups maintained the improvements obtained, with a slight increase in pain for the BFR intervention group in one study [29]. Perhaps this phenomenon could be due to the fact that the subjects could not practice the exercises with this tool once the intervention was over and did not benefit from the effects provided by the exercise practiced in isolation. Along the same lines, the control group improved to a greater extent than the BFR intervention group, with a total benefit of 8.6 points on the KOOS [28]. In the Myositis Damage Index (MDI), the BFR intervention group improved 22.3 points in global patient damage, the control group improved 17 points for global physical damage and, in the global damage section, there was no change for any of the groups [28]. The same was shown for the Kujala Patellofemoral pain, with both groups improving slightly by up to four points [34]. This variation in results may be due to several factors. First, the BFR intervention group improved the global damage section due to the strength exercises. Second, the control group improved the overall physical injury possibly due to a regression to the mean. Finally, none of the groups improved in the global damage section, possibly because there was no progression in the pressures applies by the BFR intervention [45].

### 4.4. Quality of Life

With regard to the quality of life variable, in the short-term, changes were observed in each of the areas of the Health Questionnaire SF-36, with 10 points in the physical function area, 13 points in bodily pain, foour points in vitality, 13 points in social function, 18 points in emotional role, and 11 points in mental health for the control group whereas the BFR intervention group improved 29 points in the physical role section and six points in general health [38]. However, both groups improved similarly in all sections, which may indicate that aerobic exercise with or without BFR intervention produces the same impact on the subjects’ quality of life, although the application of below-average pressures may have influenced the benefits presented by the BFR intervention, such as metabolic stress [2]. Along the same lines, the Multiple Sclerosis Impact Scale-29 (MSIS-29) showed differences in favor of the control group, especially in the motor area, with a change of 10 points, and, in the psychological area, both groups improved equally by three points [38]. For the Veterans RAND 12-Item Health Survey (VR-12), the BFR intervention group showed changes in the physical area of 15.44 points and 19 points in the mental area [43]. These data may have to do with the fact that the BFR intervention group added strength exercises to the routine of the control group including the use of high pressure with the BFR intervention cuff, which may allow for hypoalgesia and exercise with less pain.

In the medium-term, mean differences in the SF-36 of between 4 and 11 points for the mental area, 10 and 13 points in the physical area, 28 points in the physical role, almost 13 points in bodily pain, more than five points in general health, and around eight points in emotional role were found for the BFR intervention group. In the control group, improvements of almost 10 points were observed in the vitality and social role [33,40]. These results show that all exercise modalities are very interesting options to improve the functional and psychological area of the patient with knee pathology. In addition, improvements of 16.7 points on the WOMAC scale were reported for Experimental group 2 [33]. The use of low-load exercise without BFR was slightly superior to the group using BFR intervention; however, this was not statistically significant.

Although no long-term results were found, short-term data were collected at follow-up, varying slightly from the time of intervention in the case of the SF-36 [28,38], and they were maintained for the study using the MSIS-29. This may suggest that the benefits of the BFR intervention are maintained in the long-term or that patient self-efficacy is high for the intervention program developed by the researchers [61]. Nonetheless, in one of the SF-36 studies, there was heterogeneity between the groups from the beginning, even when performing exercises, possibly due to the lack of transfer of the exercises to the patient’s daily life [69].

### 4.5. Physical Activity and BFR Intervention

According to the results analyzed from the different studies included in this systematic review, it can be observed how the BFR intervention acts as a therapeutic adjuvant that can be used with different objectives depending on the conditions of the execution of the intervention (including the patient, environment, therapist...). In this sense, it is important to highlight that it is an adjunctive technique to physical exercise, aimed therefore at enhancing the effects of execution in different conditions. An improvement in different functional capacities related to strength and aerobic capacity was observed (Table 3), which was comparable to that produced during physical activity performed at a higher intensity. In this sense, the BFR intervention can be used to “simulate” higher physical capacity intensities when the subject is not capable of working on his own at these intensities. On the other hand, if the subject is capable of working at high intensities, it is possible that the BFR intervention could be an adjuvant that helps to enhance the effects that physical activity causes by itself.

### 4.6. Strengths and Limitations

There are several systematic reviews focusing on BFR intervention. However, this study can be considered as one of the few to have focused on different neuro-musculoskeletal populations and on a wide adult age range, which is interesting from a clinical point of view. For example, the application of BFR intervention with low-load exercises obtains improvements in pain and functionality in the short-term with respect to high-load exercises. These results allow for the clinical consideration of BFR intervention, since the LOP used in the different studies remained between 30% and 80%, which indicates that there is a wide range where benefits are obtained in the patients who use it. Another point to highlight is the absence of adverse effects mentioned in the literature such as rhabdomyolysis and deep vein thrombosis, whose incidence is reported to be 0.07–0.2%, thus the methodologies used comply with good practice in the use of BFR intervention and would not have exceeded the workloads or the predetermined values of creatine kinase [45].

However, this study had some limitations that must be taken into account when analyzing the results obtained. In this sense, the search was carried out in eight international databases, and studies published in two different languages were included: English, the language of scientific publications worldwide, and Spanish, the authors’ native language and one of the most widely spoken globally. However, there may be some relevant studies published in another language or database that were excluded from those selected due to these criteria. In addition, future studies should take into account the age of the participants as an independent variable in the analysis of the results, seeking a greater personalization of the treatment and an improvement in the proposed interventions. In addition, it could be interesting to investigate in depth their possible effects on the physiological, biological, and subjective variables of the patient in order to analyze their relationship. On the other hand, it is important to consider that, although common variables were analyzed between the different selected studies, there was heterogeneity in the pathologies of the participants in the different studies. For this reason, it would be necessary to design and develop more studies in each of the pathologies analyzed in order to compare the results presented in this systematic review.

## 5. Conclusions

The main conclusion that can be drawn after analyzing the different studies included in this systematic review is that BFR intervention is an adjuvant to physical exercise that seems to help improve performance in both strength and aerobic training. In this sense, the observed benefits include a decrease in the pain threshold and an improvement in the functionality and quality of life of neuro-musculoskeletal patients during the first six weeks. In addition, in the short-term, it was observed that the results are better or equal than strength exercises at high loads. However, the results provided by this tool are still not clear for medium- and long-term interventions. Therefore, further studies along this line of research may find it interesting to discern whether the effects of BFR intervention are maintained or, in contrast, tend to equalize with strength exercises at high loads due to a lack of mechanical stress.

## Figures and Tables

**Figure 1 ijerph-20-01401-f001:**
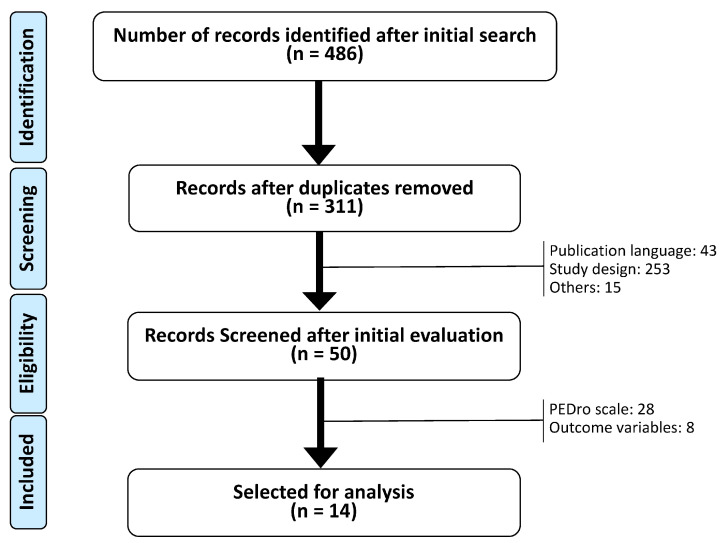
Flowchart CONSORT.

**Table 1 ijerph-20-01401-t001:** Results of the internal assessment of the documents selected according to the PEDro scale.

	Random Allocation	Concealed Allocation	Baseline Comparability	Blind Subjects	Blind Therapists	Blinding Assessors	Adequate Follow-Up	Intention to Treat	Between-Group Comparisons	Variability Outcome	Total
Branco Ferraz et al. [33]	•	-	•	•	-	-	-	•	•	•	**6**
Constantinou et al. [34]	•	•	•	-	•	•	•	-	•	•	**8**
Giles et al. [30]	•	•	•	-	•	•	•	•	•	•	**9**
Harper et al. [35]	•	-	•	•	-	-	-	•	•	•	**6**
Hughes et al. [36]	•	•	•	•	-	-	•	-	•	•	**7**
Hughes et al. [37]	•	•	•	•	-	-	•	-	•	•	**7**
Jorgensen et al. [29]	•	-	-	•	-	-	•	•	•	•	**6**
Korakakis et al. [32]	•	•	•	•	-	-	•	•	•	•	**8**
Lamberti et al. [38]	•	•	•	•	-	-	•	•	•	•	**8**
Mason et al. [39]	•	•	•	-	-	•	•	-	•	•	**7**
Rodrigues et al. [40]	•	-	•	•	-	-	•	•	•	•	**7**
Segal et al. [41]	•	•	•	•	-	-	•	-	•	•	**7**
Segal et al. [42]	•	•	•	•	-	-	•	-	•	•	**7**
Tennent et al. [43]	•	•	•	•	-	-	•	-	•	•	**7**

**Table 2 ijerph-20-01401-t002:** Characteristics of the studies included in this systematic review.

Author	Size	Age	Pressure Cuff	Interventions	Frequency Sessions	Development Interventions	Pathology
Branco Ferraz et al. [33]	n = 48 EG₁: 16 EG₂: 16 BFR: 16	EG₁: 59.9 ± 4		EG₁: High-intensity workout	20 min/ss 2 ss/w Time: 12 w	EG₁: 1 week (4 s, 10 reps, 50% 1RM), 2 week (4 s, 10 reps, 80% 1RM), 5 week (5 s, 10 reps, 80% 1RM).	Knee osteoarthritis
		EG₂: 60.7 ± 4		EG₂: Low-intensity workout		EG₂: 1 week (4 s, 15 reps, 25% 1RM), 2 week (4 s, 15 reps, 30% 1RM), 5 week (5 s, 15 reps, 30% 1RM)	
		BFR: 60.3 ± 3	70% LOP	BFR: EG₂ + BFR.		BFR: EG₂ + BFR.	
Constantinou et al. [34]	n = 60 CON: 30 BFR: 30	CON: 30.5 (18–40)		CON: High-load workout	3 ss/w Time: 4 w F/U: 8 w	CON: Hip and knee exercise program 70% 1RM, 3 s, 10 reps, 30 s rest/s. Rep: 1 s concentric–2 s eccentric.	Patellofemoral pain
		BFR: 25.5 (18–40)	70% LOP	BFR: BFR + Low-load workout		BFR: Hip and knee exercise program + BFR 30% 1RM, 4 s (reps: 30,15,15,15), 30 s rest/s, 2 min rest/ex.	
Curran et al. [31]	n = 34 EG₁: 8 EG₂: 8 BFR: 9 EG₃: 9	EG₁: 16.1 ± 2.6		EG₁: Concentrics.	2 ss/w Time: 8 w	EG₁: 1 s 20% 1RM (PC) + 4 s leg press 70% 1RM concentric–20% 1RM eccentric.	Anterior cruciate ligament reconstruction
		EG₂: 18.8 ± 3.9		EG₂: Eccentrics.		EG₂: PC+ 4 s leg press 20% 1RM concentric-70% 1RM eccentric.	
		BFR: 15.3 ± 0.9	80% LOP	BFR: Concentrics + BFR		BFR: PC + 4 s leg press 70% 1RM concentric-20% 1RM eccentric + BFR.	
		EG₃: 16.0 ± 1.7		EG₃: Eccentrics + BFR		EG₃: PC+ 4 s leg press 20% 1RM concentric-70% 1RM eccentric + BFR.	
Giles et al. [30]	n = 79 EG₁: 39 BFR: 40	EG₁: 26.7 ± 5.5		EG₁: Strength training	Trt: 3 ss/w, 8 w (6 individual ss/1–3 w) F/U: 16 w	EG1: 5 min bicycle, leg press 0–60°, y knee extension 45–90°; VAS +2/10 > ↓ 20% load (PC) + 3 s, 7–10 reps, 70% 1RM, placebo BFR (2 fingers skin/cuff).	Patellofemoral pain
		BFR: 28.5 ± 5.2	60% LOP	BFR: EG₁ + BFR		BFR: PC + 1 s (30 reps or volitive fatigue), 3 s (15 reps), 30% 1RM, 30 s rest.	
Harper et al. [35]	n = 35 EG₁: 19 BFR: 16	EG₁: 69.1 ± 7.1		EG₁: Moderate-resistance training	3 ss/w Time: 12 w	EG₁: wmup + leg press, leg extension, leg curl, and calf flexion at 60% 1RM + Flexibility–Balance ex.	Knee osteoarthritis
		BFR: 67.2 ± 5.2	Pressure mm Hg = 0.5 (SBP) + 2(thigh circumference) + 5	BFR: EG₁ + BFR		BFR: EG₁ + BFR 20% 1RM (↓ pression/s).	
Hughes et al. [36]	n = 28 EG₁: 14 BFR: 14	EG₁: 29 ± 7		EG₁: High-resistance training	2 ss/w (48 h rest/ss) Time: 8 w	EG₁: 5 min bicycle no resistance and 10 reps unilateral leg press-low load, 5 min rest (PC) + unilateral leg press 70% 1RM, 3 s, 10 reps, 30 s rest.	Anterior cruciate ligament reconstruction
		BFR: 29 ± 7	80% LOP	BFR: EG₁ + BFR		BFR: PC + EG₁ + BFR 30% 1RM, 4 s (reps: 30, 15, 15, 15).	
Hughes et al. [37]	n = 28 EG₁: 14 BFR: 14	EG₁: 29 ± 7		EG₁: High resistance training	2 ss/w (48 h rest/ss) Time: 8 w	EG₁: 5 min bicycle no resistance and 10 reps unilateral leg press-low load, 5 min rest (PC) + unilateral leg press 70% 1RM, 3 s, 10 reps, 30 s rest.	Anterior cruciate ligament reconstruction
		BFR: 29 ± 7	80% LOP	BFR: EG₁ + BFR		BFR: PC + EG₁ + BFR 30% 1RM, 4 s (reps: 30, 15, 15, 15).	
Jørgensen et al. [29]	n = 22 CON: 11 BFR: 11	CON: 69.8 ± 4.8		CON: No workout.	2 ss/w Time: 12 w F/U: 12 w	CON: Nothing.	Sporadic inclusion body myositis
		BFR: 68.1 ± 6.4	110 mm Hg	BFR: Strength training + BFR		BFR: leg press, knee extension, knee flexion (4 w), calf raise, and dorsal flexion. 3 s × 25 reps (9 w: 4 s)	
Korakakis et al. [32]	n = 40 EG₁: 20 BFR: 20	EG₁: 29.7 ± 7.6		EG₁: Low-resistance training	1 session	EG₁: Knee extension open-kinetic chain. 4 s (reps: max reps, 15, 15, 15), 30 s rest. Max load 5 kg, VAS 4/10. Rep: 2 s concentric, 2 s eccentric metronome.	Anterior knee pain
		BFR: 29.1 ± 6.6	80% LOP	BFR: EG₁ + BFR		BFR: EG₁ + BFR	
Lamberti et al. [38]	n = 22 CON: 11 BFR: 11	CON: 56 ± 10		CON: Physiotherapy assisted walking	2 ss/w Time: 6 w F/U: 6 w	CON: PC + 40 min physiotherapy assisted walking-60 m corridor. Rest: 8/10 RPE on chair.	Severe multiple sclerosis
		BFR: 54 ± 11	30% systolic blood pressure	BFR: Walking interval-metronome + BFR		BFR: 10 min wmup (PC) + 5 cycles (3 reps: 1 min work y 1 min rest. 3 min rest cycle deflated BFR) low velocity-walking (60 steps/min-metronome) + 10 min cool down and stretching CORE (PC).	
Mason et al. [39]	n = 17 CON: 9 BFR: 8	CON: 24 (20–28)		CON: Resistance exercises	2–3 ss/w Time: 12 w F/U: 12 w	CON: 4 s (reps: 30, 15, 15, 15), plus 5 lb if 75 reps in less than 5 min. Ph 1: Isometric quadriceps, and flex-ext and abd-add hip straight leg raises; Ph 2: Ph 1 + knee extension 45°–90°; Ph 3: Ph 1 + Ph 2 + hamstring curls; Ph 4: Full weight-bearing, and squats and single leg press up to 60° knee flexion.	Meniscal repair surgery
		BFR: 23 (20–26)	80% LOP	BFR: CON + BFR		BFR: CON + BFR	
Rodrigues et al. [40]	n = 48 CON: 16 EG₁: 16 BFR: 16	CON: 58.1 ± 5.9		CON: No workout	2 ss/w Time: 12 w	CON: Activities of daily living.	Rheumatoid arthritis
		EG₁: 58.0 ± 6.6		EG₁: High-load workout		EG₁: Bilateral leg press and knee extension. 1 Week: 4 s, 10 reps, 50% 1RM; 2 Week: 4 s, 10 reps, 70% 1RM; 5 Week: 5 s, 10 reps, 70% 1RM.	
		BFR: 59.6 ± 3.9	70% LOP	BFR: Low-load workout + BFR		BFR: EG₁. (1 Week: 4 s, 15 reps, 20% 1RM; 2 Week: 4 s, 15 reps, 30% 1RM; 5 Week: 5 s, 15 reps, 30% 1RM)	
Segal et al. [41]	n = 42 CON: 22 BFR: 20	CON: 56.1 ± 7.7		CON: Low-load workout	3 ss/w Time: 4 w F/U: 3 d	CON: Leg press 30% 1RM: 4 s (reps: 30, 15, 15, 15), 30 s rest. Rep: 2 s concentric and 2 s eccentric.	Knee osteoarthritis
		BFR: 58.4 ± 8.7	1 Week: 160 mm Hg 2 Week: 180 mm Hg 3 Week: 200 mm Hg	BFR: CON + BFR.		BFR: CON + BFR.	
Segal et al. [42]	n = 45 CON: 24 BFR: 21	CON: 54.6 ± 6.9		CON: Low-load workout	3 ss/w Time: 4 w F/U: 3 d	CON: Leg press 30% 1RM: 4 s (reps: 30, 15, 15, 15), 30 s rest. Rep: 2 s concentric–2 s eccentric.	Knee osteoarthritis
		BFR: 56.1 ± 5.9	1 Week: 160 mm Hg 2 Week: 180 mm Hg 3 Week: 200 mm Hg	BFR: CON + BFR.		BFR: CON + BFR.	
Tennent et al. [43]	n = 24 CON: 13 BFR: 11	CON: 37.0 (32–47)		CON: Physiotherapy	12 ss Time: 6 w	CON: Immediate weight loading, immediate formal physiotherapy and no range of motion restrictions.	Non-reconstructive knee arthroscopy
		BFR: 37.0 (30–46.2)	80% LOP	BFR: Physiotherapy + (Strength training + BFR)		BFR: CON + 4 sets (reps: 30, 15, 15, 15), 30% 1RM, 30 s rest-1 min rest/ex. (leg press, leg extension, and reverse press). 5 min max. occlusion/ex.	

BFR: blood flow restriction group; BFR_i_: blood flow restriction intermittent; BFR_c_: blood flow restriction continue; CON: control group; EG₁: Experimental group 1; EG₂: Experimental group 2; EG₃: Experimental group 3; F/U: follow-up; HR: heart rate; kg: kilogram; LOP: limb occlusion pressure; PC: common process; MVIC: maximal voluntary isometric contraction; reps: repetitions; RES: high load/traditional resistance training; RM: maxim repetition; RPE: rating perceived exertion; s: seconds; VAS: Visual Analogic Scale.

**Table 3 ijerph-20-01401-t003:** Results of the functionality variable (objective outcomes), analyzed in the short-, medium- and long-term, both during the intervention period and in a follow-up that could be longer than 6 months after the end of the intervention.

Measurement Tool		Article	Group	Baseline	Measurements (sd/ci 95%)			Follow-Up (sd/ci 95%)		
					0–6 Week	6–12 Weeks	3–6 Months	1–3 Months	3–6 Months	>6 Months
TIME	Time Up and Go Test (s)	Branco Ferraz et al. [33]	EG₁	6.75 ± 1.5 *	-	6.5 ± 1 *	-	-	-	-
			EG₂	7 ± 0.8 *	-	6.85 ± 0.7 *	-	-	-	-
			BFR	6.9 ± 0.6 *	-	6.65 ± 0.35 *	-	-	-	-
		Rodrigues et al. [40]	EG₁	7.15 ± 0.7 *	-	6.4 ± 0.6 *	-	-	-	-
			BFR	7.25 ± 1.4 *	-	6.7 ± 0.95 *	-	-	-	-
			CON	7.35 ± 1.1 *	-	7.3 ± 1.2 *	-	-	-	-
	TSA (s)	Tennent et al. [43]	BFR	9.50 (5.9–12.9)	5.11 (4.5–8.0)	-	-	-	-	-
			CON	5.84 (4.5–8.0)	4.92 (4.0–7.1)	-	-	-	-	-
	FSST	Tennent et al. [43]	BFR	7.39 (6.5–10.0)	5.89 (5.6–6.8)	-	-	-	-	-
			CON	8.45 (7.2–9.4)	6.36 (5.9–7.6)	-	-	-	-	-
	STS5 (s)	Tennent et al. [43]	BFR	10.62 (9.6–12.7)	7.77 (6.5–9.3)	-	-	-	-	-
			CON	11.27 (10.0–13.0)	7.98 (7.6–10.1)	-	-	-	-	-
		Lamberti et al. [38]	BFR	24 (8–40)	18 (7–28)	-	-	20 (5–35)	-	-
			CON	27 (1–53)	23 (3–44)	-	-	24 (2–46)	-	-
SPEED	400 m walk gait speed (m/s)	Harper et al. [35]	EG₁	1.05 *	1.05 ± 0.035 *	1.02 ± 0.045 *	-	-	-	-
			BFR	1.05 *	0.955 ± 0.03 *	1.005 ± 0.4 *	-	-	-	-
	T25FW (m/s)	Lamberti et al. [38]	BFR	0.78 (0.54–1.03)	0.90 (0.64–1.16)	-	-	0.87 (0.62–1.12)	-	-
			CON	0.76 (0.51–0.99)	0.79 (0.54–1.03)	-	-	0.76 (0.49–1.02)	-	-
	SSWV (m/s)	Tennent et al. [43]	BFR	1.31 (0.9–1.6)	1.80 (1.5–2.0)	-	-	-	-	-
			CON	1.45 (1.6–1.3)	1.91 (1.6–1.4)	-	-	-	-	-
DISTANCE	6-Min Walk Test (m)	Lamberti et al. [38]	BFR	215 (153–278)	264 (188–340)	-		266 (186–345)	-	-
			CON	183 (120–245)	218 (152–285)	-		223 (155–291)	-	-
	Modified SEBT (%LL)	Hughes et al. (2019) [36]	EG₁	-	ANT-N: 7.5 ± 8.0 ANT-I: 9.0 ± 3.5 PM-N: 8.5 ± 7.2 PM-I: 5.5 ± 5.2 PL-N: 9.8 ± 9.7 PL-I: 5.8 ± 8.0	ANT-N: 10.5 ± 9.2 ANT-I: 17.5 ± 6.7 PM-N: 12.8 ± 9.1 PM-I: 13.9 ± 7.7 PL-N: 14.5 ± 10.1 PL-I: 13.2 ± 10.3	-	-	-	-
			BFR	-	ANT-N: 8.4 ± 5.1 ANT-I: 22.3 ± 5.2 PM-N: 11.6 ± 8.1 PM-I: 19.1 ± 9.2 PL-N: 13.0 ± 15.6 PL-I: 23.3 ± 12.5	ANT-N: 18.7 ± 9.3 ANT-I: 32.9 ± 9.7 PM-N: 22.4 ± 13.7 PM-I: 32.1 ± 15.1 PL-N: 23.8 ± 17.8 PL-I: 34.8 ± 15.3	-	-	-	-
REPETITIONS	Timed Stand Test (reps)	Branco Ferraz et al. [33]	EG₁	14.25 ± 3.75 *	-	16.5 ± 4.5 *	-	-	-	-
			EG₂	13 ± 2.5 *	-	14 ± 2.5 *	-	-	-	-
			BFR	13.5 ± 2.5 *	-	15 ± 2 *	-	-	-	-
		Rodrigues et al. [40]	EG₁	13.25 ± 2.5 *	-	15.25 ± 2.65 *	-	-	-	-
			BFR	14.5 ± 3.25 *	-	16 ± 2.8 *	-	-	-	-
			CON	13.75 ± 3.75 *	-	13.5 ± 2.6 *	-	-	-	-
POWER	Stair Climb Power (W)	Segal et al. (2015) [41]	BFR	364.3 ± 71.2	-	-	-	29.3 ± 11.6 ¨	-	-
			CON	404.3 ± 118.4	-	-	-	53.4 ± 11.0 ¨	-	-

5STS: 5-time sit-to-stand; ANT-I: anterior side injured lower limb; ANT-N: anterior side non-injured lower limb; BFR: blood flow restriction group; CON: control group; EG₁: Experimental group 1; EG_2_: Experimental group 2; FSST: Four square step test; m: meters; m/s: meters per second; PL-I: posterolateral side injured lower limb; PL-N: posterolateral side non-injured lower limb; PM-I: posteromedial side injured lower limb; PM-N: Posteromedial side injured lower limb; SEBT: star excursion balance test; s: seconds; reps: repetitions; SSWV: self-selected walking velocity; STS5: sit-to-stand 5 times; TSA: timed stair ascent; T25FW: timed 25-foot walk test; %LL: leg length in percentages. * The authors show the results in figures that report an estimate value. ¨ The authors reported some changes between the baseline and measurements or follow-up. The authors did not report this information.

**Table 4 ijerph-20-01401-t004:** Functionality (subjective outcomes) analyzed in the short-, medium-, and long-term, both during the intervention period and in a follow-up that could be longer than 6 months after the end of the intervention.

Measurement Tool		Article	Group	Baseline	Measurements (sd/ci 95%)			Follow-Up (sd/ci 95%)		
					0–6 Week	6–12 Weeks	3–6 Months	1–3 Months	3–6 Months	>6 Months
FUNCTIONALITY	Short Physical Performance Battery (0–12)	Harper et al. [35]	EG₁	10.2 ± 1.9	-	0.2 ± 0.3 *	-	-	-	-
			BFR	10.4 ± 1.9	-	0.8 ± 0.5 *	-	-	-	-
	LLFDI (30–80)	Harper et al. [35]	EG₁	-	-	TF: 0.2 ± 2 * TL: 1 ± 3.75 *	-	-	-	-
			BFR	-	-	TF: -0.5 ± 1.65 * TL: 7.5 ± 2.7 *	-	-	-	-
	IKDC (0–100)	Hughes et al. [36]	EG₁	-	13.50 ± 7.42	23.33 ± 8.76	-	-	-	-
			BFR	-	22.44 ± 5.27	35.63 ± 7.06	-	-	-	-
		Curran et al. [31]	EG₁	-	-	19.98 ± 17.30	-	-	-	-
			EG₂	-	-	15.81 ± 18.02	-	-	-	-
			BFR	-	-	9.97 ± 15.96	-	-	-	-
			EG₃	-	-	13.69 ± 18.12	-	-	-	-
	LEFS (0–80)	Hughes et al.) [36]	EG₁	-	14.69 ± 7.76	21.83 ± 7.06	-	-	-	-
			BFR	-	21.46 ± 10.68	31.08 ± 12.22	-	-	-	-
		Mason et al. [39]	CON	-	-	9 ± 15 ^	-	-	19 ± 6 ^	-
			BFR	-	-	20 ± 12^	-	-	8 ± 10 ^	-
	KOOS (0–100)	Hughes et al. [36]	EG₁	-	P: 11.67 ± 6.11 S: 12.17 ± 5.91 ADL: 11.17 ± 6.28 QOL: 12.50 ± 13.85	P: 22.00 ± 7.48 S: 24.50 ± 7.62 ADL: 21.75 ± 6.90 QOL: 20.31 ± 12.82	-	-	-	-
			BFR	-	P: 30.25 ± 9.29 S: 22.17 ± 11.65 ADL: 21.83 ± 8.35 QOL: 15.10 ± 10.81	P: 39.75 ± 11.74 S: 33.33 ± 13.60 ADL: 32.33 ± 10.37 QOL: 29.58 ± 14.81	-	-	-	-
	Lysholm Knee- Scoring Scale (0–100)	Hughes et al. [36]	EG₁	-	17.25 ± 9.96	29.50 ± 12.07	-	-	-	-
			BFR	-	29.75 ± 12.86	44.58 ± 14.75	-	-	-	-
	Tegner Activity Scale (0–10)	Hughes et al. [36]	EG₁	7.42 ± 1.24	-	-	-	-	-	-
			BFR	6.83 ± 1.80	-	-	-	-	-	-
	IBMFRS (0–40)	Jørgensen et al. [29]	CON	10-I: 30.4 ± 4.4 5-I: 29.7 ± 4.9	-	10-I: - 5-I: -	-	10-I: 13.0 ± 3.4 5-I: 13.2 ± 3.3	-	-
			BFR	10-I: 31.6 ± 5.7 5-I: 14.0 ± 3.9	-	10-I: - 5-I: -	-	10-I: 32.5 ± 4.9 5-I: 14.6 ± 3.5	-	-
	HAQ (0–3)	Jørgensen et al. [29]	CON	1.05 ± 0.85	-	-	-	1.02 ± 0.79	-	-
			BFR	0.77 ± 0.58	-	-	-	0.89 ± 0.73	-	-
		Rodrigues et al. [40]	EG₁	0.38 ”	-	0.23 ”	-	-	-	-
			BFR	0.36 ”	-	0.16 ”	-	-	-	-
			CON	0.38 ”	-	0.23 ”	-	-	-	-
	BBS (0–56)	Lamberti et al. [38]	BFR	48 (43–54)	50 (45–54)	-	-	48 (42–54)	-	-
			CON	44 (39–50)	46 (39–54)	-	-	45 (37–53)	-	-

5-I: Five items; 10-I: 10 items; ADL: activities of daily living; BBS: Berg Balance Scale; BFR: blood flow restriction group; CON: control group; EG₁: Experimental group 1; EG₂: Experimental group 2; EG₃: Experimental Group 3; HAQ: Health Assessment Questionnaire; IBMFRS: Inclusion Body Myositis Functional Rating Scale; IKDC: International Knee Documentation Committee; KOOS: Knee Osteoarthritis Outcome Score; LEFS: Lower Extremity Function Scale; LLFDI: Late Life Function and Disability Instrument; P: pain; QOL: quality of life; S: symptoms; TF: total frequency; TL: total limitation. * The author shows the results in figures that report an estimate value. The authors did not report this information. ” The authors did not show the standard deviations. ^ The author included the information in general and percentages.

**Table 5 ijerph-20-01401-t005:** Pain analyzed in the short-, medium-, and long-term, both during the intervention period and in a follow-up that could be longer than 6 months after the end of the intervention.

Objective Outcomes									
Measurement Tool	Article	Group	Baseline	Measurements (sd/ci 95%)			Follow-Up (sd/ci 95%)		
				0–6 Week	6–12 Weeks	3–6 Months	1–3 Months	3–6 Months	>6 Months
KOOS (0-100)	Segal et al. [41]	BFR	83.3 ± 15.4	2.9 ± 10.0 ¨	-	-	4.9 ± 3.3 ¨^	-	-
		CON	76.6 ± 22.1	5.6 ± 11.7 ¨	-	-	14.2 ± 7.2 ¨^	-	-
	Segal et al. [42]	BFR	80.5 ± 16.9	-	-	-	2.0 ± 2.8 ¨	-	-
		CON	76.0 ± 20.0	-	-	-	1.8 ± 2.7 ¨	-	-
	Tennent et al. [43]	BFR	P: 52.8 (40.3–61.8) S: 47.10 (42.0–64.3) ADL: 58.08 (44.5–72.1) QOL: 31.3 (15.6–46.9) SP: 10.00 (0–33.75)	P: 75.0 (58.3–84.7) S: 76.8 (58.9–89.3) ADL: 88.24 (50.4–95.2) QOL: 59.34 (46.9–70.3) SP: 47.5 (37.5–71.25)	-	-	-	-	-
		CON	P: 69.40 (66.7–72.2) S: 67.90 (39.3–75) ADL: 73.50 (66.2–75.0) QOL: 43.80 (31.25–50) SP: 35.00 (10.0–45.0)	P: 77.80 (61.1–91.7) S: 71.40 (46.4–89.3) ADL: 75.00 (63.2–98.5) QOL: 62.50 (37.5–81.25) SP: 70.00 (10.0–90.0)	-	-	-	-	-
VAS (0–100 mm)	Giles et al. [41]	EG₁	WP: 51.4 ± 15.3 P-ADL: 42.5 ± 22.8	-	WP: 29.2 ± 25.6 P-ADL: 23.5 ± 24.1	-	-	WP: 25.8 ± 27.1 P-ADL: 23.9 ± 25.4	-
		BFR	WP: 55.7 ± 13.9 P-ADL: 58.2 ± 17.5	-	WP: 27.4 ± 20.1 P-ADL: 21.6 ± 25.0	-	-	WP: 28.1 ± 25.5 P-ADL: 31.7 ± 26.6	-
	Rodrigues et al. [40]	EG₁	3.22 ”	-	3.15	-	-	-	-
		BFR	4.73	-	2.30	-	-	-	-
		CON	2.59	-	2.81	-	-	-	-
DAS-28 (0–10)	Rodrigues et al. [40]	EG₁	2.76 ± 0.79	-	-	-	-	-	-
		BFR	2.72 ± 1.0	-	-	-	-	-	-
		CON	2.66 ± 0.8	-	-	-	-	-	-
Kujala Patellofemoral Score (0–100)	Giles et al. [41]	EG₁	72.6 ± 10.5	-	83.2 ± 12.3	-	-	85.9 ± 13.3	-
		BFR	73.6 ± 9.9	-	86.5 ± 10.5	-	-	84.4 ± 12.0	-
	Constantinou et al. [34]	CON	74.1 (71.66- 76.54)	94.1 (92.25–96.09)	-	-	98.7 (97.38–99.95)	-	-
		BFR	72.7 (69.89- 75.57)	94.9 (93.19–96.61)	-	-	98.9 (97.81–99.99)	-	-
MDI (0–38)	Jørgensen et al. [29]	CON	GD: 0.17 ± 0.04 PGD: 55.2 ± 17.8 PHGD: 52.8 ± 8.5	-	-	-	GD: 0.17 ± 0.07 PGD: 46.9 ± 15.7 PHGD: 35.8 ± 9.7	-	-
		BFR	GD: 0.18 ± 0.05 PGD: 48.5 ± 12.1 PHGD: 45.0 ± 18.2	-	-	-	GD: 0.19 ± 0.06 PGD: 28.3 ± 10.7 PHGD: 33.0 ± 19.0	-	-
NPRS (0–10)	Korakakis et al. [32]	CON	SLS-S: 3.8 ± 2.3 SLS-D: 5.1 ± 1.8 SDT: 4.1 ± 2.6	SLS-S: 2.6 ± 2.7/2.5 ± 2.3 SLS-D: 4.2 ± 2.2/4.0 ± 2.2 SDT: 2.2 ± -2.2/2.9 ± 2.2	-	-	-	-	-
		BFR	SLS-S: 4.6 ± 2.3 SLS-D: 5.6 ± 2.6 SDT: 4.2 ± 2.4	SLS-S: 2.0 ± 1.6/2.0 ± 1.5 SLS-D: 2.9 ± 2.3/3.7 ± 2.3 SDT: 3.0 ± 2.5/2.2 ± 2.1	-	-	-	-	-
Pain Börg Scale (0–11)	Hughes et al. [37]	EG₁	-	MP: 0.7 ± 0.4 (I)/1.6 ± 0.6 (NI) * KP: 2.6 ± 1 (I)/3.3 ± 1 (NI) *	MP: 0.8 ± 0.4 (I)/0.9 ± 0.5 (NI) * KP: 1.3 ± 0.95 (I)/0.3 ± 0.3(NI) *	-	-	-	-
		BFR	-	MP: 4 ± 1.5 (I)/4.75 ± 1 (NI) * KP: 0.4 ± 0.5 (I)/0.3 ± 0.2 (NI) *	MP: 3.75 ± 1.5 (I)/4.3 ± 0.9 (NI) * KP: 0.1 ± 0.2 (I)/0.05 ± 0.1 (NI) *	-	-	-	-
WOMAC Pain Subscale (0–20)	Harper et al. [35]	EG₁	7.23 ± 4.87	-	0.3 ± 1.4	-	-	-	-
		BFR	6.19 ± 3.04	-	0.9 ± 1.05	-	-	-	-

ADL: activities of daily living; BFR: blood flow restriction group; CON: control group; DAS-28: Disease Activity Score; EG₁: Experimental group 1; GD: global damage; KOOS: Knee Osteoarthritis Outcome Score; KP: knee pain; MDI: Myositis Damage Index; MP: muscle pain; P: pain; P-ADL: pain-activities of daily living; PGD: patient global damage; PHGD: physical global damage; QOL: quality of life; S: symptoms; SDT: side-down test; SLS-S: single leg squat shallow; SLS-D: single leg squat deep; SP: sport; VAS: Visual Analogic Scale; WOMAC: Western Ontario and McMaster Universities Osteoarthritis Index; WP: worst pain experiments last week. * The author shows the results in figures that report an estimate value. The authors did not report about this information. ” The authors did not show the standard deviations. ¨ The authors reported some changes between the baseline and measurements or follow-up. ^ The authors included the information in general and percentages.

**Table 6 ijerph-20-01401-t006:** Quality of life analyzed in the short-, medium-, and long-term, both during the intervention period and in a follow-up that may be longer than 6 months after the end of the intervention.

Measurement Tool	Article	Group	Baseline	Measurements (sd/ci 95%)			Follow-Up (sd/ci 95%)		
				0–6 Week	6–12 Weeks	3–6 Months	1–3 Months	3–6 Months	>6 Months
SF-36 (0–100)	Branco Ferraz et al. [33]	EG₁	MH: 65.4 ± 20.7 PH: 55.7 ± 16.9	-	MH: 71.1 ± 23.1 PH: 64.8 ± 15.5	-	-	-	-
		EG₂	MH: 69.0 ± 15.7 PH: 57.0 ± 15.9	-	MH: 78.5 ± 19.8 PH: 66.0 ± 20.3	-	-	-	-
		BFR	MH: 68.0 ± 23.8 PH: 60.4 ± 16.1	-	MH: 79.3 ± 12.0 PH: 73.4 ± 13.5	-	-	-	-
	Jørgensen et al. [29]	CON	36.4 ± 21.7	-	-	-	32.3 ± 20.4	-	-
		BFR	54.5 ± 11.4	-	-	-	57.8 ± 17.6	-	-
	Rodrigues et al. [40]	EG₁	PHF: 73.44 ” RPH: 85.71 ” BP: 68.94 ” GH: 57.50 ” V: 71.56 ” SF: 83.75 ” RE: 83.81 ” MH: 74.75 ”	-	PHF: 83.67 ” RPH: 100.0 ” BP: 70.73 ” GH: 62.53 ” V: 81.33 ” SF: 93.40 ” RE: 88.87 ” MH: 79.73 ”	-	-	-	-
		BFR	PHF: 73.13 ” RPH: 60.71 ” BP: 56.44 ” GH: 51.25 ” V: 69.06 ” SF: 86.81 ” RE: 83.38 ” MH: 77.75 ”	-	PHF: 83.33 ” RPH: 88.46 ” BP: 69.13 ” GH: 56.53 ” V: 75.33 ” SF: 91.80 ” RE: 91.07 ” MH: 81.33 ”	-	-	-	-
		CON	PHF: 72.81 ” RPH: 71.43 ” BP: 72.75 ” GH: 60.38 ” V: 73.44 ” SF: 82.88 ” RE: 87.50 ” MH: 79.00 ”	-	PHF: 77.33 ” RPH: 82.69 ” BP: 71.20 ” GH: 57.93 ” V: 76.33 ” SF: 83.47 ” RE: 80.00 ” MH: 77.87 ”	-	-	-	-
	Lamberti et al. [38]	BFR	PHF: 43 (31–54) RPH: 57 (41–72) BP: 60 (41–79) GH: 37 (26–47) V: 53 (44–62) SF: 54 (41–66) RE: 70 (44–95) MH: 64 (49–79)	PHF: 48 (36–60) RPH: 86 (66–107) BP: 66 (45–87) GH: 43 (29–58) V: 54 (44–65) SF: 64 (51–76) RE: 76 (53–98) MH: 63 (51–75)	-	-	PHF: 45 (32–59) RPH: 70 (49–91) BP: 64 (41–87) GH: 36 (26–46) V: 52 (43–60) SF: 64 (48–80) RE: 82 (59–105) MH: 67 (55–80)	-	-
		CON	PHF: 36 (21–51) RPH: 56 (34–78) BP: 62 (41–84) GH: 40 (33–47) V: 46 (38–54) SF: 49 (37–61) RE: 73 (53–92) MH: 66 (54–77)	PHF: 46 (29–64) RPH: 84 (62–106) BP: 75 (59–92) GH: 44 (33–54) V: 50 (39–60) SF: 62 (49–74) RE: 91 (81–101) MH: 75 (63–87)	-	-	PHF: 43 (27–59) RPH: 77 (53–101) BP: 75 (59–93) GH: 40 (30–50) V: 49 (37–61) SF: 60 (45–76) RE: 88 (73–103) MH: 71 (55–85)	-	-
MSIS-29 (0–100)	Lamberti et al. [38]	BFR	m: 62 (51–72) p: 24 (19–29)	m: 58 (48–68) p: 21 (16–26)	-	-	m: 57 (46–67) p: 22 (16–27)	-	-
		CON	m: 61 (51–71) p: 21 (17–25)	m: 51 (42–61) p: 18 (14–22)	-	-	m: 53 (42–65) p: 19 (14–23)	-	-
VR-12 (0–100)	Tennent et al. [43]	BFR	PCS: 30.86 (22.4–39.4) MCS: 51.20 (41.2–59.5)	PCS: 46.3 (38.2–52.1) MCS: 60.24 (55.5–63.9)	-	-	-	-	-
		CON	PCS: 36.50 (25.3–40.1) MCS: 57.60 (54.2–63.9)	PCS: 47.70 (35.6–50.5) MCS: 56.20 (50.4–61.5)	-	-		-	-
WOMAC (0–98)	Branco Ferraz et al. [33]	EG₁	36.6 ± 11.1	-	21.2 ± 13.2	-	-	-	-
		EG₂	35.1 ± 16.2	-	18.4 ± 11.5	-	-	-	-
		BFR	31.5 ± 12.0	-	17.1 ± 11.2	-	-	-	-

BFR: blood flow restriction group; BP: bodily pain; EG₁: Experimental group 1; EG₂: Experimental group 2; GH: general health; m: motor component; MCS: Mental Component Score; MH: mental health; MSIS-29: Multiple Sclerosis Impact Scale-29; p: psychological component; PCS: Physical Component Score; PFD: physical function domain; PH: physical health; PHF: physical function; RE: role emotional; RPH: role physical; SF: social function; SF-36: Health Questionnaire SF-36; V: vitality; VR-12: Veterans RAND 12-Item Health Survey; WOMAC: Western Ontario and McMaster Universities Osteoarthritis Index. ” The authors did not show the standard deviations.

## Data Availability

Data are available under request.

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
