# Peer review of "Effectiveness of Blood Flow Restriction on Functionality, Quality of Life and Pain in Patients with Neuromusculoskeletal Pathologies: A Systematic Review"

_ijerph, 2023, doi:10.3390/ijerph20021401_

Round 1

Reviewer 1 Report

I really appreciate the opportunity to review this manuscript entitled “Effectiveness of blood flow restriction on functionality, quality of life and pain in patients with neuromusculoskeletal pathologies: A systematic review.” This is important to assess the effectiveness of emerging treatments in this population.  I remark some issues (most of them in methods) in order to improve the quality of this manuscript.

The abstract is clear but it is important that keywords summarize the paper, as a suggestion a key word related to neurosmusculoskeletal disorders should be included. Introduction was well structure and shows the necessity for this research. The aim of the paper is clear at the end of the introduction.  

At the methods section, there are some questions that should be review. About exclusion criteria, why do you choose PEDro scale? This is an external evaluation that is already down in PEDro database, authors should use another tool to make their own critical assessment.  

About results is the same question as methods. Table 1 should be redone with another critical assessment tool. Discussion summarize and explain in a good way the finding. Conclusions were correct but can be more concise.

Reviewer 2 Report

Effectiveness of blood flow restriction on functionality, quality of life and pain in patients with neuromusculoskeletal pathologies: A systematic review.

 General comments

This is a paper on BFR and different outcomes for neuromusculoskeletal pathologies.

The article is good but more robust justification is needed for the introduction, results interpretation and discussion.

There is some methodological concerns to address.

Figure 1 is missing.

Introduction

The introduction is well written, straight to the point.

Though, the auteurs may consider adding those points to the introduction:

-        Could it be appropriate at some point to make a distinction between blood flow restriction and remote ischemic preconditioning? Ex.: https://pubmed.ncbi.nlm.nih.gov/31333890/

-        There are many reviews on BFR, you may cite them better by explaining to the lector what they would add to their comprehension of BFR (at some point in relation with your review) . Ex.: https://pubmed.ncbi.nlm.nih.gov/28259850/, https://pubmed.ncbi.nlm.nih.gov/33196300/ , https://pubmed.ncbi.nlm.nih.gov/30306467/, https://pubmed.ncbi.nlm.nih.gov/33375515/, https://pubmed.ncbi.nlm.nih.gov/25249278/, https://pubmed.ncbi.nlm.nih.gov/29043659/, https://pubmed.ncbi.nlm.nih.gov/29043659/

-        With you results, conclusion and discussion, you put an emphasis on long term effect of BFR, this notion should be discussed in introduction as well. Data were collected at the baseline, short (0-6 weeks), medium (6-12w) and long-term 109

(12-24w), the follow-up period as the post-treatment period: What is the rational?

Line 56. Nonetheless, the number of reviews that exist today is insufficient, and they tend to evaluate specific clinical populations or a particular variable. I understand there is no review but what about clinical trials? Maybe cite an example from those single trials to get your point.

Line 58: On the other hand, it can be said that there is a gap in the literature. What is precisely the gap, and why it is important?

Methods

The words ischemic training was used as a keyword, how do you differentiate from ischemic preconditioning?

Line 73: research up to 2021 ……which month? How come not until now (near 2023) ?  A from when?

Functional: Did you encounter any specific functional tests such as SFMA, sit and stand, 6MWT or other?

Questionnaires of Quality of life: do you have example of which one may, be used or you accept anything? Did you accept only validated questionnaire?

Line 104: You state disease, but you work targets MSKI, adjust accordingly.

It is not clear in your method (inclusion criteria) what is the population you target, and in results this segment provides confusion (ex.: From your abstract: Objective: The main objective of this systematic literature review was to analyze the effects BFR on pain, functionality and quality of life in subjects with neuromusculoskeletal pathologies…. It should be mentioned on how you confirmed it was a neuromusculoskeletal pathologies.

Add more exclusion criteria: ex.: master thesis, etc.

I don’t get the point of all those languages, in all the article you have selected how many are truly in French, Portuguese, Italian or Spanish?

Results

What is the total N of your review?

From your abstract: Objective: The main objective of this systematic literature review was to analyze the effects BFR on pain, functionality and quality of life in subjects with neuromusculoskeletal pathologies…. therefore, I don’t fully understand why Cardoso et al. (2019) paper was included if the patients are End-stage Renal Disease? I think you should focus on 1 pathology (ex.: the most recurrent for knee).

Line 116: Where is Figure 1.? I think it is missing….

You include both cardio and strength…. good, but more rational is needed (maybe in intro) to justify this choice.

For age group, do you think it may be relevant to separate different kind of age group?

Discussion

In general, the discussion looks more of a reporting than a serious discussion on how to really apply BFR for the targeted patients.

There is no clear discussion on the potential effect of BFR vs the exercise intervention.

It is not clear in the discussion (for all data reported) if the hypothesis/explanation come from the author or the cited article. There should be a clear distinction. Ex.: Line 306`For the pain outcome, short-term changes of 5.6 points are shown for the control group in the KOOS [31]. Perhaps the lack of specificity in applying the necessary pressure to occlude the artery is not enough to create adaptations in the subject, as well as to work on progression [47]. Try to report clearly what the study reported in their own! Then, you may argue with other reference, etc…

Line 172: This systematic review analyzing the effects of BFR and different exercise methodologies show changes in favor of BFR in the short term for each of the aforementioned variables despite the great heterogeneity of the population groups. You did not perform a meta-analysis so you cannot say it show changes…furthermore it is not clear which help the most between exercises methodology and BFR.

Line 334: Pain Börg Scale: Analog pain scale and Borg scale are 2 different things. A Borg scale is to quantify cardiovascula exercise…not pain!

For all your results, I think you should discuss if they are clinically relevant vs statistically relevant. Example: Is it clinically relevant to increase of 49 meters on the 6MWT for those patients (https://erj.ersjournals.com/content/37/1/150 )? What does the improvement mean from a health perspective? From a normative perspective?  All gain seems little (from a clinical perspective) are not meaning full for non-athlete population unless it really has a major clinical outcome. The auteur should discuss more on this for each result presented.

I am not sure to understand the rational on how BFR could increase the quality of life of an individual. In my opinion, this is an example of wanting to push to much on the biopsychosocial model and to much on the psychosocial side, leaving aside the bio.

Line 387: BFR tool…it is not a tool, but an intervention.

Limits: All the pathology are not exactly the same, you need to do some cluster to draw conclusion.

Line 395: on a wide adult age range, which is interesting from a clinical point of view. You may find it interesting, but I don’t agree it is a strength! Furthermore, I have some concern about teenagers (do they got a consent form appropriate from their age) that are in the same analysis as older people. I don’t think that all age group respond the same? I may be mistaken, but I have a doubt! What about responder’s vs non responders?

The language part is irrelevant, English is the common scientific language.

Conclusion

Line 412: The data collected in this review indicate that the BFR tool is a therapeutic alternative due to its effectiveness under different exercise modalities...You must change your conclusion: 1rst: you don’t have data, this is not a meta-analysis. 2nd: An alternative to what????? It more of a adjuvant than an alternative?
